# SEARCHING FOR PARAMETER-EFFICIENT TUNING ARCHITECTURE FOR TEXT-TO-IMAGE DIFFUSION MODELS

## ABSTRACT

The large-scale text-to-image diffusion model, represented by Stable Diffusion, has achieved remarkable success in the field of image generation. Transferring pretrained diffusion models to downstream domains with parameter-efficient tuning (PEFT) methods such as Adapter and LoRa have become the most common paradigms. Despite their widespread usage, there has been limited research on systematically studying how the design of these components would impact the final tuning effectiveness. In this paper, we investigate the automatic design of an optimal tuning architecture. Specifically, we employ a reinforcement learning-based neural network search method to facilitate the automatic design of the tuning architecture for PEFT of Stable Diffusion with few-shot training data. Our search space includes micro-structures similar to Adapter, LoRa, as well as their insertion positions. For effective searching and evaluation, we build a large-scale tuning dataset. Through our search, we successfully obtained a novel tuning architecture that reduces parameter count by 18% compared to the widely adopted LoRa approach but still surpasses across various downstream tasks hugely. We also conduct extensive analysis of the searched results, aiming to provide valuable insights to the community regarding parameter-efficient tuning for large-scale diffusion models.

## 1 INTRODUCTION

Recently, diffusion models have achieved remarkable success in various generative tasks, such as image generation Zhang & Agrawala (2023); Mou et al. (2023); Rombach et al. (2022); Radford et al. (2021); Saharia et al. (2022), 3D generation Xu et al. (2023); Poole et al. (2022); Jain et al. (2022), image inpainting Xie et al. (2023); Lugmayr et al. (2022), and video generation Harvey et al. (2022); Ho et al. (2022), due to their high-fidelity and high-diversity generation capability.

Among these applications, text-to-image generation Radford et al. (2021); Saharia et al. (2022); Rombach et al. (2022) is particularly popular. Users can input a textual description, known as a prompt, and the diffusion model can generate high-quality images corresponding to the prompt. Stable Diffusion Rombach et al. (2022) is currently the most popular open-sourced model. It is trained on LAION-5B dataset Schuhmann et al. (2022), which contains 5 billion text-image pairs, enabling it to provide a comprehensive depiction of objects. However, the performance of Stable Diffusion in generating images for specific domains is not satisfactory. Therefore, transfer learning, which effectively adapts a publicly available and large-scale pretrained Stable Diffusion model to a specific domain, has become a popular application paradigm.

The pioneering work of transferring pretrained diffusion models to downstream tasks is Dreambooth Ruiz et al. (2023), which tunes all parameters of the diffusion model to adapt to the specific object of interest. However, diffusion models often have a large number of parameters and incur significant training and inference costs due to their multi-step denoising process. For example, Stable Diffusion adopts a U-Net as its denoiser. Although it first uses a VAE to map images to a latent space, the U-Net still contains approximately 861 million parameters, resulting in significant costs for full fine-tuning. Therefore, reducing the cost of fine-tuning large-scale pretrained diffusion models has become a huge challenge. Inspired by the parameter-efficient tuning (PEFT) methods first

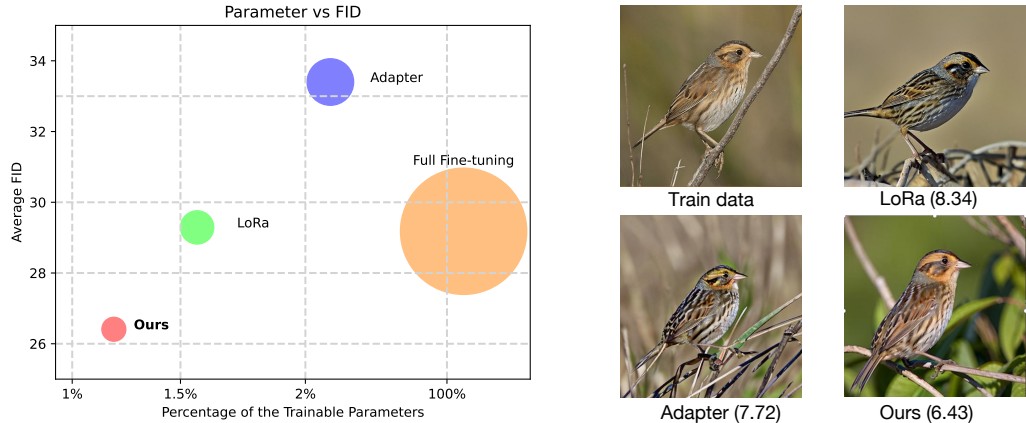

Figure 1: **Left**: Average FID score on downstream tasks *vs.* trainable parameters. Our method achieves better transferring results while tuning the fewest parameters. **right**: Examples of tuning results on Nelson's sparrow from CUB-200-2011 dataeset. Our searched architecture achieves better performance for both FID ($\downarrow$) and visual quality.

investigated in natural language processing (NLP), such as Adapter He et al. (2022) and LoRa Hu et al. (2021) and prefix tuning Li & Liang (2021), the community tries to conduct PEFT on Stable Diffusion. Currently, facilitated by the open-source WebUI, the community usually directly applies LoRa for tuning, which adds low-rank bypasses to simulate full fine-tuning. However, compared to the simple transformer structure commonly used in NLP, the U-Net in Stable Diffusion has more components, including residual blocks, self-attention, cross-attention, FFN, as well as different stages like downsampling, bottleneck, and upsampling. Therefore, there is more room for designing parameter-efficient tuning architecture for the Stable Diffusion.

In this paper, we aim to investigate automatic neural architecture design of parameter-efficient tuning methods for Stable Diffusion He et al. (2022) with reinforcement learning-based methods Zoph & Le (2016); Tan et al. (2019) We want to determine what plug-in structures are most suitable and where they should be inserted into the Stable Diffusion U-Net, as well as how they affect the final results. Therefore, our search space is set to the micro-structure design of plug-in modules like Adapter and LoRa, as well as their insertion locations. The are many sub-modules and connections in Stable Diffusion UNet where we can inject our searched plug-in modules, making our search space super flexible. For searching and evaluation, we sample images from existing fine-grained recognition datasets which consist of many sub-classes, and collect images from the internet to build a large-scale text-conditioned image tuning benchmark. Through a sample-train-evaluate loop based on reinforcement learning and maximizing rewards, we discovered that better tuning protocols tend to inject Adapter-like module after the cross-attention block and add LoRa-like module for both the cross-attention and FFN block. As shown in Figure 1, compared to the widely used LoRa, our method reduces the parameter number by 18% while achieving a 2.88 decrease in FID score for generation performance and demonstrating better visual quality. Compared to full fine-tuning, we decrease the FID score by 2.77 while tuning only 1.29% of the full parameters. We also validate the generalization and transferability of the discovered structures on a large number of data domains.

## 2 BACKGROUND

### 2.1 RELATED WORKS

#### 2.1.1 PARAMETER-EFFICIENT FINE TUNING

The concept of parameter-efficient fine-tuning Houlsby et al. (2019); Hu et al. (2021); Li & Liang (2021); Lester et al. (2021); Jia et al. (2022) was initially introduced by natural language process-

ing. With the advent of super large-scale models Scao et al. (2022); Thoppilan et al. (2022) like BERT Devlin et al. (2018) and GPT Brown et al. (2020); Radford et al. (2018), these models often possess a vast number of parameters, presenting challenges for downstream transfer learning. Performing a full tuning requires substantial GPU memory and computational resources, leading to extensive research efforts aimed at reducing the cost. These efforts primarily focus on two types of methods.

Some methods involve introducing new structures, such as Adapter Houlsby et al. (2019) and LoRa Hu et al. (2021). Adapter are commonly inserted between transformer layers and comprise a down-projection layer, a non-linear activation function, and an up-projection layer. While LoRa emphasizes low-rankness. And some works Hyeon-Woo et al. (2021); Valipour et al. (2022); Liu et al. (2022) improve the architecture of LoRa. Other methods are known as prefix tuning Li & Liang (2021); Lester et al. (2021); Jia et al. (2022), which add learnable parameters before inputs or activations while maintaining the original model structure.

Parameter-efficient fine-tuning has achieved remarkable success in NLP and has also become the popular approach for tuning large-scale vision models Sung et al. (2022); Chen et al. (2022).

### 2.1.2 TEXT-TO-IMAGE DIFFUSION MODEL

In recent years, text-to-image has made significant progress. Early work mostly relied on GANs Crowson et al. (2022); Esser et al. (2021); Reed et al. (2016); Tao et al. (2022) to generate images, while the diffusion model Ho et al. (2020); Nichol & Dhariwal (2021) has been recently proposed. The diffusion model relies on training a denoising autoencoder to learn the inverse of a Markovian diffusion process. A large number of diffusion-based models Mou et al. (2023); Zhang & Agrawala (2023) such as Dalle Ramesh et al. (2021),Imagen Saharia et al. (2022) and Stable Diffusion Rombach et al. (2022) have greatly improved the effectiveness of text-to-image generation. The most widely used model is Stable Diffusion, which includes a CLIP text encoder, a VAE structure for compressing images into a latent space, and a conditional UNet for learning the diffusion process from noise to image. However, the parameter of Stable Diffusion is huge, and transferring it to the downstream domain will introduce huge computation costs. Dreambooth Ruiz et al. (2023) uses full tuning. While applying LoRa which is supported by open-source WebUI is the most common paradigm. In addition, evaluating the quality of generated images is also an important problem in text-to-image generation. Recently, Wu et al. (2023) proposed an approach that provides a scoring model for automated evaluation of text-conditioned image generation by mimicking human preference.

### 2.2 CONDITIONAL DIFFUSION MODEL

Let's begin by briefly exploring the diffusion model. Diffusion models involve applying Gaussian noise to the initial data $x_0$ through a series of data corruptions. The corruption process can be defined as follows, for a given timestep $t = 1, \ldots, T$, where $T$ indicates the number of noise levels and $\beta_t$ represents the noise scale:

$$q(x_t|x_0) = \mathcal{N}(\sqrt{\bar{\alpha}_t}x_0, (1 - \bar{\alpha}_t)\mathbf{I}) \tag{1}$$

In the above equation, $\bar{\alpha}_t$ is a hyperparameter derived from the $\beta_t$ values, calculated as $\bar{\alpha}_t = \prod_{s=1}^{t} \alpha_s$. The main goal of diffusion models is to recover the original data $x_0$ from the corrupted data $x_T$ by reversing the corruption process. DDPM Ho et al. (2020) proposes a Markov chain to gradually denoise $x_{T...1}$ through transitions: $p_\theta(x_{t-1}|x_t) = \mathcal{N}(\mu_\theta(x_t, t), \Sigma_\theta(x_t, t))$. The covariance $\Sigma_\theta(x_t, t)$ is fixed as hyperparameters and in DDPM, while the $\mu_\theta$ is parameterized as a noise prediction network $\epsilon_\theta$. The training objective of Ho et al. (2020) can be represented by :

$$\mathcal{L} = \mathbb{E}_{x_0,\epsilon,t}\left[|\epsilon - \epsilon_\theta(x_t(x_0, \epsilon), t; \mathcal{C})|_2^2\right] \tag{2}$$

Here, $\mathcal{C}$ represents the conditioning inputs. Currently, the most popular diffusion model is Stable Diffusion Rombach et al. (2022), which applies a diffusion process in latent space.

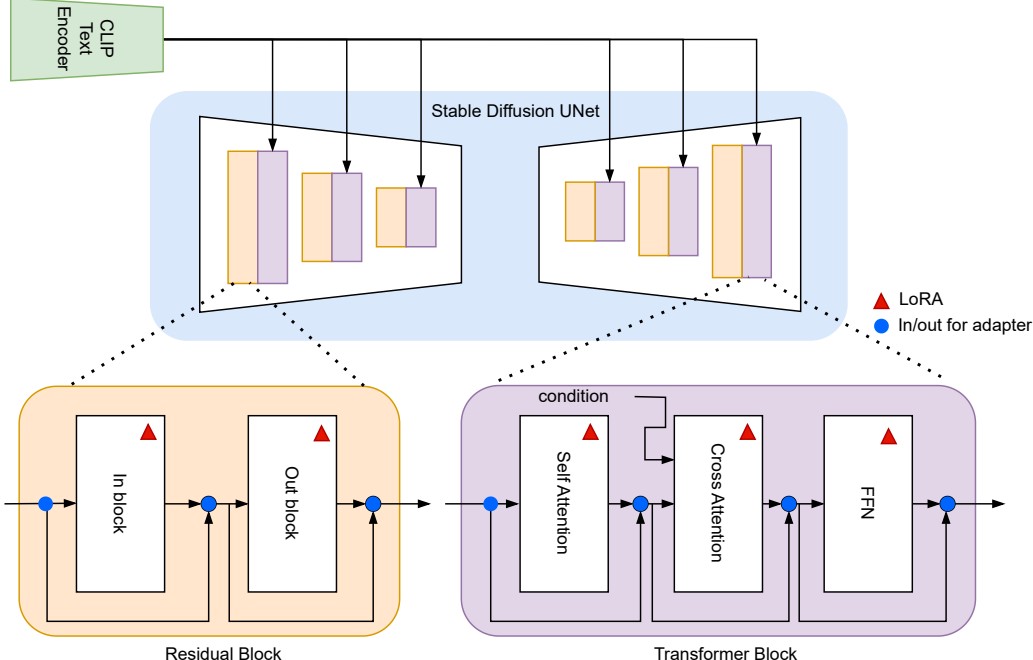

Figure 2: Illustration of the overall architecture of Stable Diffusion UNet and our search space. The top shows the architecture of Stable Diffusion UNet, which consists of a stack of residual blocks and transformer blocks, as well as many stages including downsampling, bottomneck, and upsampling. The below shows possible positions for us to inject plug-in tuning modules. Blue circles indicate potential input/output positions for the Adapter-like module and red triangles represent locations where LoRa-like module can be added.

## 3 METHOD

To conduct the automatic design of parameter-efficient tuning architecture for Stable Diffusion, there are three main components for searching: search space, search objective, and search algorithm. In section 3.1, we will first introduce our carefully designed search space for efficient tuning protocol. Then, in section 3.2, we will demonstrate our search objective through a comprehensive evaluation and our search algorithm using reinforcement learning.

### 3.1 PARAMETER-EFFICIENT TUNING SEARCH SPACE

In our work, we mainly focus on designing new plug-in structures for parameter-efficient tuning for Stable Diffusion UNet. We follow two types of designs, namely Adapter Houlsby et al. (2019) and LoRa. We will also investigate where they should be inserted.

**General architecture for Stable Diffusion UNet.** As shown in the Figure 2, the Stable Diffusion UNet consist of many sub-modules. It has three stages, including downsampling, bottleneck block, and upsampling. Each stage has a stack of blocks, consisting of residual blocks and transformer blocks. The transformer block includes self-attention, cross-attention, and feedforward network (FFN). The U-Net has a total of 26 layers, and different layers are generally believed to be responsible for modeling different levels of detail Zhang & Agrawala (2023); Mou et al. (2023). Thus we want not only to search for how to design optimal Adapter-like or LoRa-like plug-in modules but also determine where they should be injected.

**Search space for Adapter-like module.** For Adapter, they are often injected between specific layers of the network and consist of a down-projection layer, a non-linear activation function, and an up-projection layer. To inject Adapter into the UNet, we carefully explore their input position and

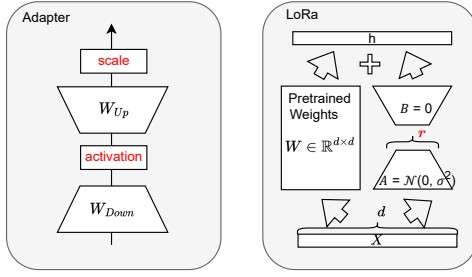
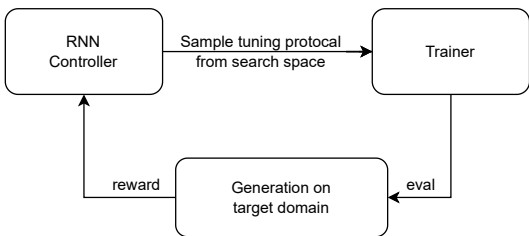

Figure 3: The search space for Adapter and LoRa-like module. For Adapter-like, we search for the activation function, scaling factor, and bottleneck dim. For LoRa-like, we simply search for adaptive rank. Please note that all the structure parameters may vary for different blocks in the Stable Diffusion UNet.

Figure 4: The overall search pipeline of our reinforcement learning-based search pipeline. We search for the best parameter-efficient tuning protocol in a sample-eval-update loop.

output position. As shown in Figure 2, each connection (denoted as blue points) can serve as the input position or output position of Adapter-like module. There are over 20 possible combinations for different in/out positions. For the structure design of the Adapter-like module, as shown in Figure 3, we focus on the choice of activation functions, including ReLU, sigmoid, Swish, Silu, and identity, as well as scale factors $s$ within the module and projection intermediate dimensions $d$.

**Search space for LoRa-like module.** Unlike Adapter, LoRa adds bypasses to weights and performs dimension reduction and expansion operations to simulate full fine-tuning. Its structure is shown in Figure 3. To apply LoRa, we mainly study which modules it should be added to in the UNet. As shown in Figure 2, all red triangular represent possible added modules. In the original LoRa, it could only be added to linear layers. In our work, we also consider whether to add it to convolutional layers as part of the search space. For the architecture design, the rank dimension $r$ of LoRa plays a critical role in its effectiveness, as a larger dimension introduces more parameters but often leads to better tuning results. Thus we search for different ranks for different modules among different stages.

Considering the above design aspects, including structure design and proper inject or add position, the size of our search space is approximately $10^{12}$, posing great challenges for traditional methods such as grid search. Therefore, an efficient search method is necessary to explore the parameter-efficient tuning protocol.

### 3.2 SEARCH OBJECTIVE AND ALGORITHM

We propose to use reinforcement learning-based search algorithms to search for the best protocol for parameter-efficient tuning with the search space defined in Section 3.1. The Recurrent Neural Network (RNN) is adopted as the controller Tan et al. (2019), generating the search parameters from our search space. Through a sample-evaluate-update loop, the parameters of the RNN are optimized to maximize the search reward. The overall search process is illustrated in Figure 4. We consider two metrics as our optimization targets: evaluation performance and the number of tuned parameters. Our goal is to achieve good tuning results while makes the tuning computation cost affordable. The final reward function is formulated as follows,

$$\mathrm{R}(\boldsymbol{C}) = \mathrm{Eval}(\boldsymbol{C}) \times \left[\frac{\mathrm{Param}(\boldsymbol{C})}{\mathrm{TAR}}\right]^{\alpha} \tag{3}$$

,

where $C$ represents the sampled combination of designs for tuning, $\mathrm{Eval}(\boldsymbol{C})$ denotes the evaluation performance of the tuned Stable Diffusion on the target domain. In our search, we apply a comprehensive evaluation metric, which is a combination of FID Heusel et al. (2017) and human preference score Wu et al. (2023). For specific domains, we will also add domain-specific evaluation metrics, such as cosine similarity between training and generated face images for tuning on faces.

$\mathrm{Param}(\boldsymbol{C})$ is the tuning parameter number of the combination. TAR is the target tuning parameter count for searching. In our search, we set it to the trainable parameter numbers of traditional LoRa (with rank of 32). We want to achieve good performance while keeping similar same tuning cost. As for $\alpha$, it is the weight factor.

We use Proximal Policy Optimization (PPO) to optimize the RNN controller for finding Pareto optimal solutions. For each sampled combination in the search space, we map it to a list of tokens, which are determined by a sequence of actions $a_{1:T}$ from the RL agent with policy $\pi_\theta$. The overall objective is to maximize the expected reward:

$$\mathbb{J} = \mathbb{E}_{P_{(a_{1:T};m)}} \mathrm{R}(\boldsymbol{C}) \tag{4}$$

Along with the sample-evaluate-update loop, the evaluation result of the searched tuning protocol tends to converge and we finally get the optimal tuning protocol for Stable Diffusion.

## 4 EXPERIMENTS

### 4.1 IMPLEMENTATION DETAILS

**Base model.** We search for the best tuning protocol for Stable Diffusion Rombach et al. (2022), which is currently the most widely used text-to-image diffusion model. We used the official release of `sd-v1-5` as our base model. The input size for images is $512 \times 512$. To evaluate the generalization of the searched tuning protocol, we also conducted validation on community models such as Chilloutmix.

**Tasks.** We consider two tasks, namely the Dreambooth task and the fine-tuning task. The Dreambooth task aims to personalize the model with few images. It consists of two types of data: personalization data and regularization data. The personalization data usually contains specific objects that the user wants to embed, typically only a few images. Its input prompt is "a photo of [V] $C_{\text{class}}$" where [V] represents the rare token and $C_{\text{class}}$ represents the class word. As for the regularization data, it consists of photos generated by the model for regularization purposes, and its prompt is "a photo of $C_{\text{class}}$." The fine-tuning task involves tuning the model on a small number of text-image pairs. We select a wide range of fine-grained recognition datasets, including Oxford 102 Flower Nilsback & Zisserman (2008), Food101 Bossard et al. (2014), SUN397 Xiao et al. (2010), Caltech101 Li et al. (2022), CUB-200 Wah et al. (2011), ArtBench Liao et al. (2022), and Stanford Cars Krause et al. (2013). We sample 50 images from each sub-class for these fine-grained datasets. Additionally, we also collected 20 styles of anime-style photos from the internet, with 50 photos per style. To evaluate the performance of fine-tuning on faces, we also collected 10 identities from the CelebA dataset, with 50 face images per identity. Totally, we get a large-scale text-to-image tuning dataset, which consists of over 13000 images from 250 sub-classes.

**Tuning and sampling**. We used AdamW as our optimizer and set the learning rate to 1e-5. For each domain's data, we train for 2.5k iterations. As for the comparison baseline LoRA, we set its rank to 32. It accounts for 1.58% of the total parameters. After model tuning, we adopt DPM-Solver++ Lu et al. (2022) as the sampling algorithm. The number of sampling steps is set to 50, and the cfg Ho & Salimans (2022) scale is set to 7.0. We keep the text encoder of Stable Diffusion fixed when tuning. For the searching process, we search for 1000 samples.

**Evaluation.** We use four metrics for evaluation: FID, human preference score, CLIP similarity score, and face similarity. FID Heusel et al. (2017) measures the feature distribution distance between generated images and original images, which is the most widely used metric for evaluating generated images. However, it cannot solve the overfitting problem. Human preference score Wu et al. (2023) evaluates the results by making a ResNet model mimicking human preferences, which is more in line with human tastes. We also use CLIP similarity scores to evaluate the Dreambooth task. We calculate the cosine similarity of CLIP features between generated images and personalized datasets. For the tuning on face data, we directly extract face features using a face recognition model Deng et al. (2019) and calculate the average cosine distance between the sampled face images and the training images. In the search phase, we set the relative weights of FID and human performance score to 2:1 as the $\mathrm{Eval}(\boldsymbol{C})$ in Eq. 3

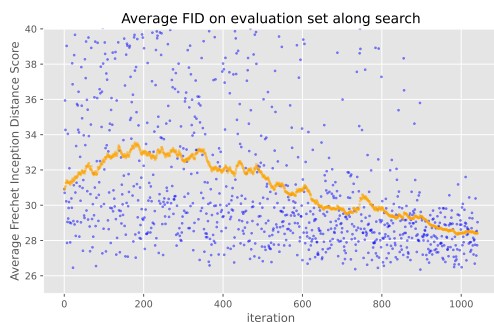
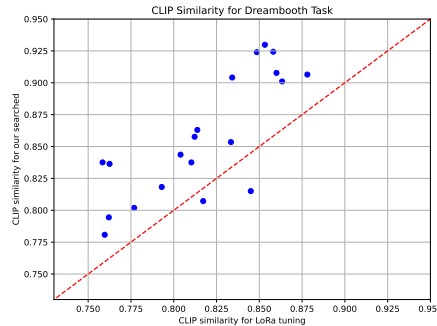

Figure 5: The average FID score along the search process. As the search progresses, FID decreases and gradually converges. The yellow line represents the moving average.

Figure 6: Tuning results comparison with our searched protocol and LoRa on Dreambooth tasks. We measure the CLIP similarity score for generated images and training personalization data.

| Method | Food101 | CUB-200 | ArtBench | Flowers102 | Anime | Average FID | Params(M) | Training Time |
|---|---|---|---|---|---|---|---|---|
| Full-tuning | 10.13 | 7.06 | 51.47 | 26.01 | 51.24 | 29.18 | 861.03 (100%) | 1× |
| Adapter | 10.78 | 7.69 | 59.27 | 26.88 | 62.42 | 33.40 | 19.85 (2.31%) | 0.49× |
| LoRa | 12.01 | 8.21 | 47.69 | 29.69 | 48.87 | 29.29 | 13.57 (1.58%) | 0.52× |
| Ours | **9.51** | **6.63** | **43.87** | **25.34** | **46.72** | **26.41** | 11.13 (1.29%) | 0.47× |

Table 1: Comparsion on fine-tuning tasks with our searched tuning protocol and existing parameter-efficient tuning methods for downstream datasets. FID score, trainable parameter, and training time are reported.

## 4.2 RESULTS

### 4.2.1 SEARCH PROCESS AND OPTIMAL PROTOCOL

In Figure 5, we show the changes in FID along with our search. As the search progresses, FID gradually decreases and ultimately converges, showing the effectiveness of our reinforcement learning-based search approach.

Throughout the search process, we found that the optimal tuning protocol tends to add an Adapter-like module after the cross-attention layer and utilize silu as the activation function. Furthermore, the optimal tuning protocol applies the LoRa-like module to cross-attention layers and FFN layers. We also observe that superior tuning results tend to add higher-rank LoRa in the upsampling stage. This is because, in contrast to the preceding two stages, namely downsampling and the bottleneck, the upsampling stage better captures object details, and thus needs a higher-rank LoRa-like module for fitting.

### 4.2.2 QUANTITATIVE RESULTS

We first present the results of Dreambooth tasks. We compare our searched protocol with LoRa. For each Dreambooth task, we measure the average CLIP feature similarity between generated images and personalization data. As shown in Figure 6, our searched method outperforms existing LoRa on most tasks, with a decrease in trainable parameters for 18%.

In Table 1, we compare our searched protocol with current popular parameter-efficient tuning methods on fine-tuning tasks. We select several domains from our evaluation benchmark and report the FID score. From the results, we can see that our method consistently outperforms competitors across all tasks. Remarkably, our approach achieves a lower FID than full tuning while only introducing 1.29% trainable parameters. We also reduced the tuning time to 47% of that required for full tuning. When compared to the widely adopted Adapter and LoRa methods, our method not only reduces

| Method | HPS score | | | | Cosine Similarity |
| --- | --- | --- | --- | --- | --- |
| | Animation | Art | Photo | Average | Face |
| Full tuning | 27.66 | 27.15 | 27.64 | 27.48 | 0.901 |
| Adapter | 27.57 | 26.96 | 27.59 | 27.37 | 0.882 |
| LoRa | 27.76 | 27.31 | 27.53 | 27.53 | 0.896 |
| Ours | **27.83** | **27.37** | **27.69** | **27.63** | **0.914** |

Table 2: Comparsion with our searched tuning protocol with the existing parameter-efficient tuning methods with human preference score Wu et al. (2023). We merge all fine-grained datasets into 'Photo' to align with the original HPS setting. As for tuning on the face, we report the average feature cosine similarity for generated face images and training sets for 20 identities.

| Searched with | Base model | FID | HPS |
| --- | --- | --- | --- |
| sd-v1-5 | sd-v1-5 | 26.41 | 27.63 |
| Chilloutmix | sd-v1-5 | 26.83 | 27.60 |
| LoRa | sd-v1-5 | 28.29 | 27.53 |

| Searched on | Target domain | FID |
| --- | --- | --- |
| Flowers | Flowers | 25.34 |
| CUB-200 Birds | Flowers | 25.62 |
| LoRa | Flowers | 29.69 |

(a) Transfer results on base model searched with. We transfer the best tuning protocol searched with Chilloutmix to `sd-v1-5`. The FID and HPS are averaged among all domains of our evaluation benchmark.

(b) Transfer results on domain data searched on. We transfer the best tuning protocol searched on CUB-200-Birds to Flowers. The FID is reported in Flowers.

Table 3: Results of transferability. We validate the transferability for our searched tuning protocol on both search data and base model. The last line for both substables means raw LoRa with rank 32 as the baseline method for comparison.

FID significantly but also decreases the trainable parameters hugely. By comparing different tuning methods, we find that the Adapter-like inject module performs better in realistic scenarios, while the LoRa-like added module performs better in artistic and abstract styles.

Furthermore, we also report the results of our searched protocol compared to other tuning methods in terms of human preference scores, as shown in Table 2. Our method consistently scores higher, indicating a better alignment with human preferences. For face generation tasks, we present cosine similarity scores in Table 2. We achieve an improvement of 0.018 compared to LoRa. This improvement suggests that our searched tuning protocol better captures facial identity features and possesses better generalization ability.

### 4.2.3 TRANSFERABILITY RESULTS

To validate the transferability of the searched protocol, we conduct experiments on data domain transfer and base model transfer. To avoid a strong coupling between the searched protocol and the data domain, we validate the best protocol obtained from searching with CUB-200-Birds on Oxford 102 Flowers. Furthermore, to avoid overfitting the tuning protocol to the base model, we also transfer the best protocol obtained from searched with Chilloutmix to `sd-v1-5`. As shown in Table 3, the transferred protocol still achieves better tuning performance compared to LoRa, demonstrating the good generalization of our searched protocol. Compared to the original domain or base model searched with, there is a slight performance drop due to differences in distributions between different models or data domain. However, compared to re-executing the search for each new base model or domain, transferring the existing searched protocol is the most cost-effective approach.

### 4.3 VISUALIZATION

In Figure 7, we present the image generation results obtained by Stable Diffusion using our searched parameter efficient tuning protocol. We showcase highly realistic details and diversity. Additional visual results are available in the appendix for a comprehensive overview.

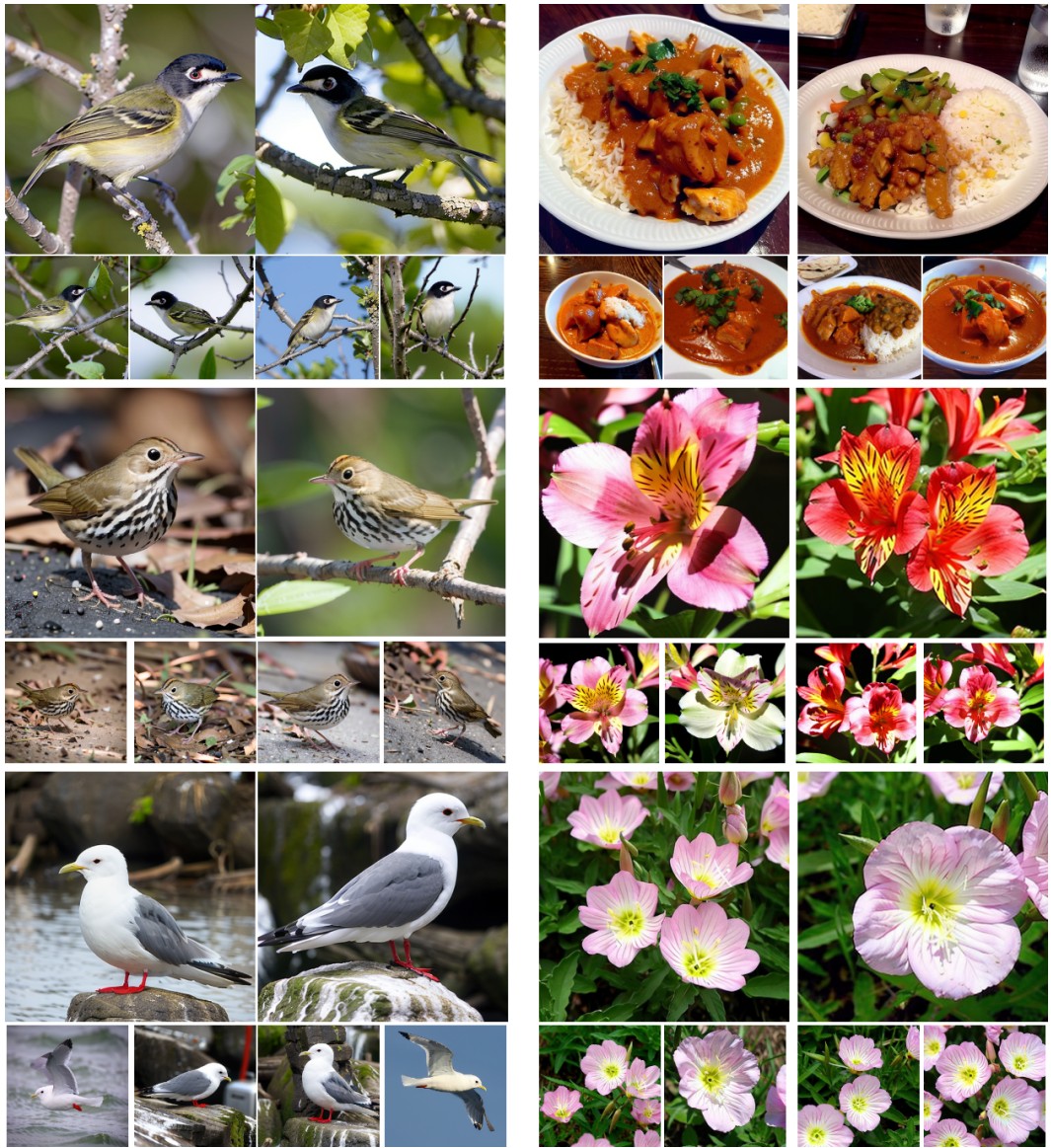

Figure 7: Visualization of image samples generated with Stable Diffusion tuned by our searched parameter-efficient protocol. We select six classes, *black-capped vireo, chicken curry, ovenbird, peruvian lily, pink primrose,* and *red legged kittiwake.* The text prompt is "a photo of $C_{\text{class}}$ ". The sampler is DPM++ with 50 steps, and the CFG scale is 7.

## 5  CONCLUSION

This paper investigates automatic neural architecture design for parameter-efficient tuning of Stable Diffusion for text-to-image generation. Through a reinforcement learning-based search, we discover a novel tuning structure that reduces parameter count by 18% while decreasing the FID score to 2.88 compared to LoRa. This approach achieves significant efficiency gains, tuning only 1.29% of model parameters. We also demonstrate the generalization of these structures across various data domains, offering valuable insights for the research community and advancing the field of large-scale diffusion model tuning in generative tasks. In future work, we will apply the search-based methods to find the optimal tuning protocol for other works, like NLP or audio.

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

# A APPENDIX

In the appendix, we provide more generation visualization results of our searched protocol.

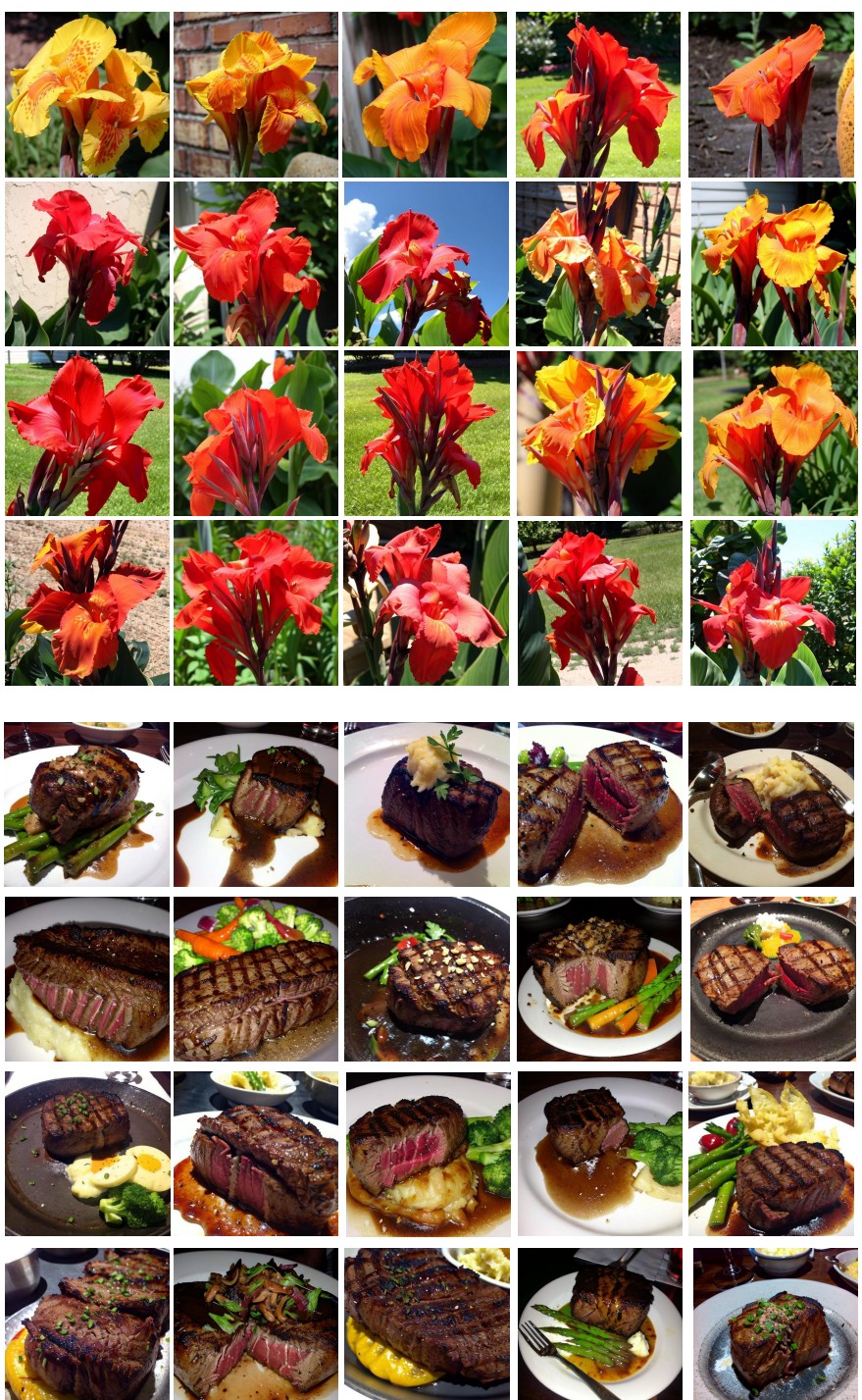

Figure 8: Visualization of image generation results on "canna lily" and "filet mignon". Steps: 50, Sampler: DPM++, CFG scale: 7.

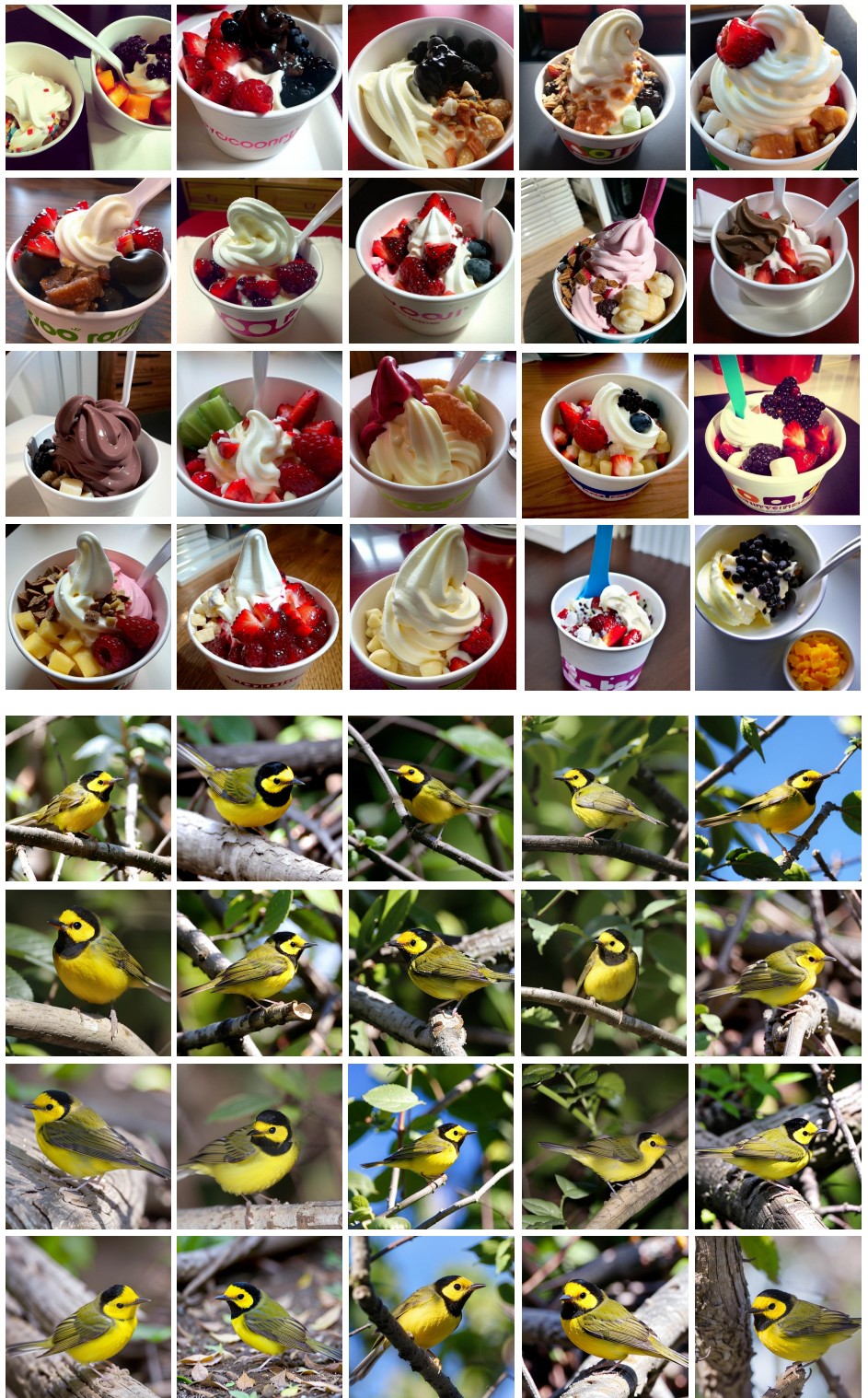

Figure 9: Visualization of image generation results on "frozen yogurt" and "hooded warbler". Steps: 50, Sampler: DPM++, CFG scale: 7.

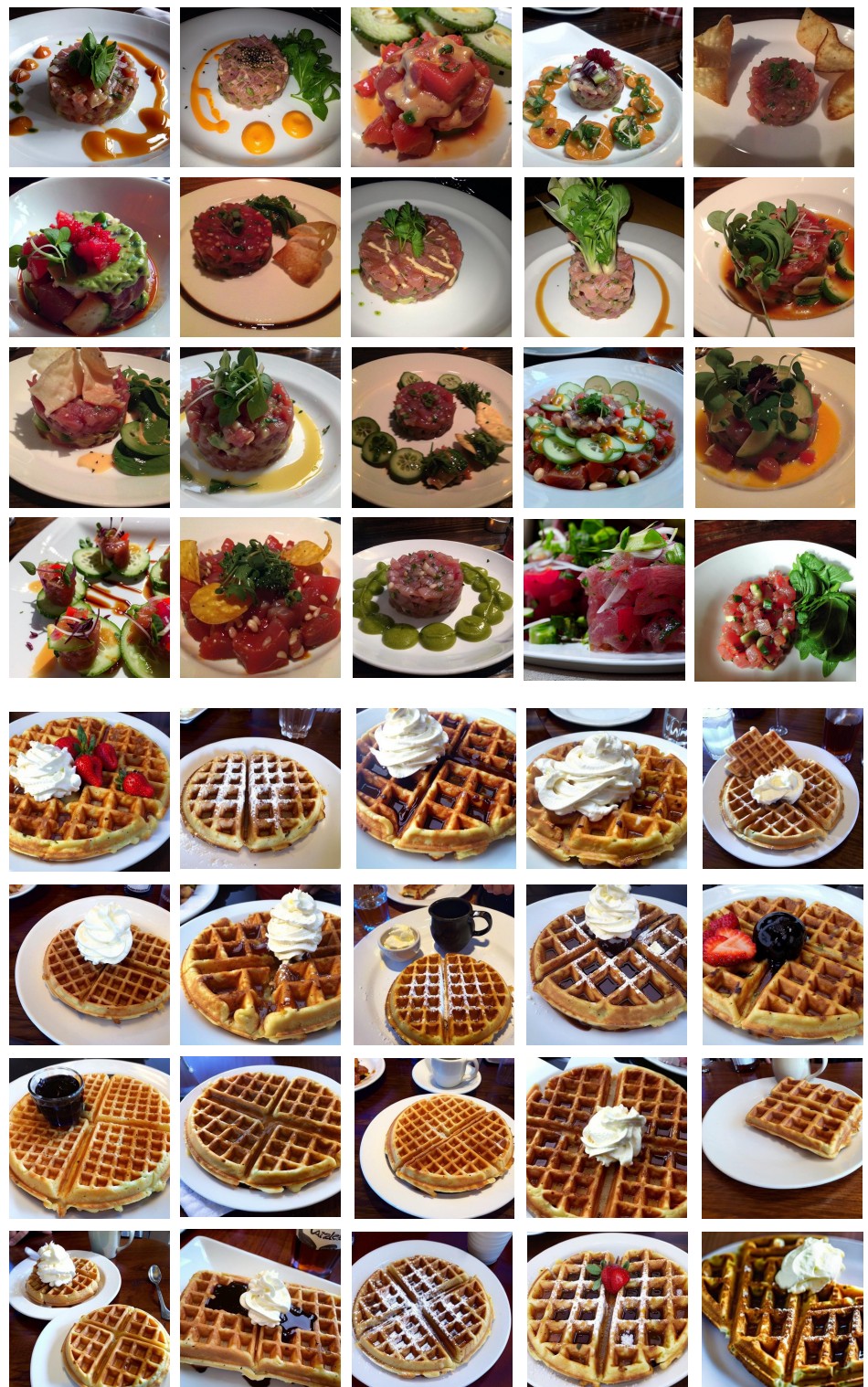

Figure 10: Visualization of image generation results on "tuna tartare" and "waffles". Steps: 50, Sampler: DPM++, CFG scale: 7.

