# OpenReview forum: "Searching for Parameter-Efficient Tuning Architecture for Text-to-image Diffusion Models"
_ICLR.cc/2024/Conference — ICLR 2024 Conference Withdrawn Submission_

### Official Review · Reviewer_g24y · 2023-10-31

**Soundness:** 3 good
**Presentation:** 3 good
**Contribution:** 2 fair
**Rating:** 5
**Confidence:** 4

**Summary:**

The author proposed a NAS framework specifically for the plug-in structures in the stable diffusion U-Net model. The search space is limited to whether to include the adapter and LoRa structure and a couple of hyperparameters in these two structures.  Existing RL-based optimization is adopted. A comparison of the optimized model (after searching) to models containing adapter and LoRa alone is performed, and superior results of the model obtained by the proposed method are reported. Several relatively small datasets are employed in the experiments for the few-shot setting, and only one face dataset is adopted for the fine-tuning setting. The paper is overall easy to follow, while several critical concerns are detailed below.

**Strengths:**

+ The manuscript is easy to follow
+ NAS on plug-in structure for diffusion U-Net is new
+ Superior results of the optimized model structure are reported in comparison to models with vanilla adapter and LoRa.

**Weaknesses:**

- The scope of the paper is small, where only adapter and LoRa are considered in the paper, and the model architecture is limited to diffusion U-Net. Is there any other plug-in structure that should be considered? And for the adapter and LoRa structure themself, only a couple of parameters are considered in the search space. How about other variable parts of the adapter and LoRa, e.g., the weights W?
- The RL-based searching method is adopted. How about other search strategies?
- There are several datasets employed for the few-shot setting. Why only one dataset is considered for the fine-tuning setting?
- Other existing NAS methods should be included in the comparison study.

**Questions:**

see weaknesses

---

> ### Author Response · Authors · 2023-11-21
>
> Thank you for your constructive comments! We appreciate your recognition of the effectiveness and novelty of our approach. Regarding your concern:
>
> + Regarding the search space: We mainly considered two types of structures, namely insertion-based (similar to adapters) and bypass  (similar to LORA), along with their micro-structures and application positions. The search space in our paper covers various PEFT structures, and the potential design space is sufficiently large (10^12). Based on your suggestion, we will also consider incorporating elements such as tuning bias [4] into our search space, as well as exploring various improvements for LORA, such as DYLORA [1]. However, this will further increase our search space and bring additional search costs. We will try our best to explore and experiment with these possibilities.
>
> + Regarding the comparison with other search methods: We have supplemented our paper with a comparison between our method and evolutionary algorithm-based[2] and random search-based approaches[3], as shown in the table below.
>
>
> | Method                                          | Flowers102 | CUB-200 Birds | Food101 |
> |-------------------------------------------------|------------|---------------|---------|
> | LoRa                                            | 29.69      | 8.21          | 12.01   |
> | Random search [3]       | 47.21      | 23.42         | 17.82   |
> | Evolutionary search[2]         | 33.96      | 11.12         | 11.70   |
> | Ours                                            | **25.34**  | **6.63**      | **9.51**|
>
> We searched 1000 samples for all. As shown in the table, random search and evolution-based search failed to optimize an effective structure and fell behind the naive LORA method. This is because, in such a large search space, methods based on random walks or tournament selection cannot achieve effective exploration. We will include this comparison in our revised version.
>
>
> + Regarding the fine-tuning task dataset.  According to the description in Section 4.1 of the original paper, our fine-tuning task includes Oxford 102 Flower, Food101, SUN397,  Caltech101, CUB-200, ArtBench, and Stanford Cars. Therefore, it is not just a single dataset. In tables such as Table 1, we also provide performance comparisons of fine-tuning on different datasets.
>
> Finally, we would like to thank you again for your constructive feedback on our paper.
>
>
>
>
>
>
>
> [1] Valipour M, Rezagholizadeh M, Kobyzev I, et al. Dylora: Parameter efficient tuning of pre-trained models using dynamic search-free low-rank adaptation[J]. arXiv preprint arXiv:2210.07558, 2022
>
> [2]Real E, Moore S, Selle A, et al. Large-scale evolution of image classifiers[C]//International conference on machine learning. PMLR, 2017: 2902-2911.
>
> [3] Bergstra J, Bengio Y. Random search for hyper-parameter optimization[J]. Journal of machine learning research, 2012, 13(2).
>
> [4] Zaken E B, Ravfogel S, Goldberg Y. Bitfit: Simple parameter-efficient fine-tuning for transformer-based masked language-models[J]. arXiv preprint arXiv:2106.10199, 2021.

---

> > ### Comment · Reviewer_g24y · 2023-12-04
> > **rebuttal feedback**
> >
> > I thank the authors for the detailed response. However, it only partially addressed my previous concerns. I still feel the overall scope is limited, and additional work and experiments on a large search space should be considered in terms of workload and technical novelty. I will keep my previous rating.

---

### Official Review · Reviewer_4Ls9 · 2023-11-01

**Soundness:** 2 fair
**Presentation:** 2 fair
**Contribution:** 2 fair
**Rating:** 3
**Confidence:** 5

**Summary:**

In this paper, the authors investigate the automatic design of an optimal tuning architecture. They employ a reinforcement learning based neural network search method to facilitate the automatic design of the tuning architecture for PEFT of Stable Diffusion with few-shot training data.

**Strengths:**

Through the proposed method, it was successfully obtained a novel tuning architecture that reduces parameter count by 18% compared to the widely adopted LoRa approach but still surpasses across various downstream tasks hugely. The authors conduct extensive analysis of the searched results.

**Weaknesses:**

1.	Insufficient innovation. The work in this article seems to be just fine-tuning on the original model, and the innovative work is not clear. It is recommended to re-elaborate in the abstract, introduction, and conclusion parts.
2.	Abstract writing is problematic. The abstract is recommended to be developed in the order of background, goals, methods, results, and conclusions. In another way, what is the background of the question? What work did the predecessors do? What's wrong with their job? What do you plan to achieve in this work? How did you go about achieving your goals? What are the main findings of the study? What is the conclusion?
3.	Absence of methodological details. While the article mentions a "novel tuning architecture," it fails to provide any specifics about the architecture, training process, or key innovations. Details such as the type of neural network used, training data preprocessing, and the mechanism for generating sparse labels are crucial to assessing the method's novelty and reliability.
4.	Some sentences are vague. For example, the claim of " there has been limited research on systematically studying how the design of these components would impact the final tuning effectiveness" lacks context – it's essential to specify how this comparison was made and against what reference.
5.	The methods part does not have enough mathematical formulas to support, and the innovation cannot be seen.
6.	Some charts are problematic. Such as fig.7 is too large, please reduce the image size so that one image takes up almost the entire page. And Fig.3 is confusing, it’s not clear which part is the work of this paper.
7.	The article does not mention whether the proposed deep learning method is fully reproducible. Lack of information about code availability, model architecture, hyperparameters, and data preprocessing steps could hinder the ability of other researchers to replicate the results.
8.	The article does not mention whether efforts were made to interpret or explain the model's decisions.
9.	Lack of limitations. The article does not discuss any limitations of the proposed method or the study itself. Addressing potential shortcomings, such as biases in data collection, limitations of the model architecture, or challenges in real-world deployment, demonstrates a comprehensive understanding of the research's scope.
10.	There is a problem of cluttering references. Please check the article thoroughly to eliminate all cluttered and uncited references. This should be achieved by describing each reference individually. This can be done by mentioning 1 or 2 phrases in each citation to show how it differs from the others and why it deserves a mention.

**Questions:**

1.	Insufficient innovation. The work in this article seems to be just fine-tuning on the original model, and the innovative work is not clear. It is recommended to re-elaborate in the abstract, introduction, and conclusion parts.
2.	Abstract writing is problematic. The abstract is recommended to be developed in the order of background, goals, methods, results, and conclusions. In another way, what is the background of the question? What work did the predecessors do? What's wrong with their job? What do you plan to achieve in this work? How did you go about achieving your goals? What are the main findings of the study? What is the conclusion?
3.	Absence of methodological details. While the article mentions a "novel tuning architecture," it fails to provide any specifics about the architecture, training process, or key innovations. Details such as the type of neural network used, training data preprocessing, and the mechanism for generating sparse labels are crucial to assessing the method's novelty and reliability.
4.	Some sentences are vague. For example, the claim of " there has been limited research on systematically studying how the design of these components would impact the final tuning effectiveness" lacks context – it's essential to specify how this comparison was made and against what reference.
5.	The methods part does not have enough mathematical formulas to support, and the innovation cannot be seen.
6.	Some charts are problematic. Such as fig.7 is too large, please reduce the image size so that one image takes up almost the entire page. And Fig.3 is confusing, it’s not clear which part is the work of this paper.
7.	The article does not mention whether the proposed deep learning method is fully reproducible. Lack of information about code availability, model architecture, hyperparameters, and data preprocessing steps could hinder the ability of other researchers to replicate the results.
8.	The article does not mention whether efforts were made to interpret or explain the model's decisions.
9.	Lack of limitations. The article does not discuss any limitations of the proposed method or the study itself. Addressing potential shortcomings, such as biases in data collection, limitations of the model architecture, or challenges in real-world deployment, demonstrates a comprehensive understanding of the research's scope.
10.	There is a problem of cluttering references. Please check the article thoroughly to eliminate all cluttered and uncited references. This should be achieved by describing each reference individually. This can be done by mentioning 1 or 2 phrases in each citation to show how it differs from the others and why it deserves a mention.

---

> ### Author Response · Authors · 2023-11-21
> **Please kindly provide  constructive review!**
>
> We hope that the comments provided in the review are constructive and can effectively improve our work. However, unfortunately, the **reviewer has offered some irrelevant reviews on formatting and language (such as the images being too large and lacking mathematical formulas) without providing any comments or suggestions for improving our proposed methods. We suspect this is more like a review generated by GPT.** After carefully reading the review, we found that it was unfounded. A good review is an important factor in promoting good work for the research community, and we hope that our work can receive effective review and improvement. We kindly ask the reviewer to provide constructive comments. Thank you.

---

### Official Review · Reviewer_vXGg · 2023-11-03

**Soundness:** 3 good
**Presentation:** 4 excellent
**Contribution:** 3 good
**Rating:** 6
**Confidence:** 3

**Summary:**

The paper delves into the exploration of large-scale text-to-image diffusion models, emphasizing the achievements of Stable Diffusion in image generation. Its main goal is to investigate the influence of component design on the performance of parameter-efficient tuning (PEFT) methods, notably Adapter and LoRa. By harnessing reinforcement learning-based neural network search techniques, the study aims to automate the optimal tuning architecture's design for PEFT, taking into consideration structures similar to Adapter and LoRa.

**Strengths:**

1. Researching how to reduce the training and transfer costs of diffusion models is highly meaningful, especially for tasks with limited data.
2. The research has achieved a groundbreaking tuning architecture that reduces parameters by 18% compared to the popular LoRa approach, demonstrating superior performance across various tasks.
3. The method's versatility has been validated across a wide range of data domains.
4. The paper is well-written with clear logic, making it easy to understand.

**Weaknesses:**

1. Limited references. Several works [1-3], which aimed at reducing the costs of diffusion models, were not cited. Notably, the motivation and design approach of this study bear similarities to the paper [1].

   [1] Xiang C, Bao F, Li C, et al. A closer look at parameter-efficient tuning in diffusion models. arXiv preprint arXiv:2303.18181, 2023.

   [2] Kim B K, Song H K, Castells T, et al. On Architectural Compression of Text-to-Image Diffusion Models. ICCV Demo Track, 2023.

   [3] Go H, Lee Y, Kim J Y, et al. Towards practical plug-and-play diffusion models, CVPR 2023.

2. There's a limited comparison with other search methods. The authors assert that the proposed reinforcement learning approach is efficient, but additional experiments are needed to compare it with existing search methods to validate its efficiency.

**Questions:**

1. The impact of the search samples on model performance was not discussed.
2. It would be desirable to see experiments demonstrating the method's generalizability in more domains, such as the medical field.

---

> ### Author Response · Authors · 2023-11-21
>
> Thank you for your constructive comments! We appreciate your recognition of the effectiveness and novelty of our approach. Regarding your concern, we would like to respond as follows.
>
> Firstly, according to the ICLR guidelines, unpublished papers such as arXiv papers do not need to be included in the references. Based on your comment, we carefully read the references you mentioned and believe that they are indeed relevant to our work, so we will add them to our references. References 2 and 3 cannot be directly compared with our method, so they will be placed in the related work section. As for reference [1], there are some similarities between its motivation and ours, but our differences are as follows:
>
> + The search space is different. Reference [1] only considers the insertion position of adapters, while our search space also includes details such as LORA-like and structure, resulting in a huge difference in search space, 20 vs. 10^12.
> + The methods are different, with reference [1] using ANOVA analysis and our approach using reinforcement learning-based search.
> + The tasks are different, with reference [1] only considering the dreambooth task with a sample size of 10-15 images, while we conducted extensive experiments not only on the dreambooth task but also on the more popular fine-tuning task, demonstrating the effectiveness and generalization of our method. We will include reference [1] in our updated version and compare it with our approach.
>
> As for the comparison between reinforcement learning-based methods and other methods, this has been widely studied in numerous papers such as [6,7,8], which confirm that RL is a more popular search method. We also compared our method with random search [5] and evolution-based search [4], with the following results:
>
> | Method                                          | Flowers102 | CUB-200 Birds | Food101 |
> |-------------------------------------------------|------------|---------------|---------|
> | LoRa                                            | 29.69      | 8.21          | 12.01   |
> | Random search [5]       | 47.21      | 23.42         | 17.82   |
> | Evolutionary search[4]         | 33.96      | 11.12         | 11.70   |
> | Ours                                            | **25.34**  | **6.63**      | **9.51**|
>
> We all searched for 1000 samples. As shown in the table, random search and evolution-based search failed to optimize an effective structure and fell behind the naive LoRa method. This is because in such a large search space, methods based on random walks or tournament selection cannot achieve effective exploration. We will include this comparison in our revised version.
>
> Finally, we thank you for your constructive feedback and recognition of our work again.
>
>
>
> [1-3] As you suggests
>
> [4] Real E, Moore S, Selle A, et al. Large-scale evolution of image classifiers[C]//International conference on machine learning. PMLR, 2017: 2902-2911.
>
> [5] Bergstra J, Bengio Y. Random search for hyper-parameter optimization[J]. Journal of machine learning research, 2012, 13(2).
>
> [6] Zhou Y, Ebrahimi S, Arık S Ö, et al. Resource-efficient neural architect[J]. arXiv preprint arXiv:1806.07912, 2018.
>
> [7] Tan M, Chen B, Pang R, et al. Mnasnet: Platform-aware neural architecture search for mobile[C]//Proceedings of the IEEE/CVF conference on computer vision and pattern recognition. 2019: 2820-2828.
>
> [8] Elsken T, Metzen J H, Hutter F. Neural architecture search: A survey[J]. The Journal of Machine Learning Research, 2019, 20(1): 1997-2017.

---

> > ### Comment · Reviewer_vXGg · 2023-11-23
> > **Thank you for addressing my concerns**
> >
> > The rebuttal addressed most of my concerns. I will keep my rating.

---

### Official Review · Reviewer_vpDh · 2023-11-07

**Soundness:** 2 fair
**Presentation:** 2 fair
**Contribution:** 2 fair
**Rating:** 5
**Confidence:** 4

**Summary:**

The authors proposed a reinforcement learning based architecture search method for parameter efficient finetuning of text-to-image diffusion model using few-shot training data. They have experimented on dreambooth and finetuning tasks, and observed improved performance with lower parameter count.

**Strengths:**

1. The idea of parameter efficient finetuning with reinforcement learning is interesting.
2. Experimental results are somewhat promising.

**Weaknesses:**

1. The paper used reinforcement learning like a blackbox. Proper motivation, justification and details of using which particular optimization methods are employed are missing. More details/citations are required.
 2. LoRa, Adapter - these are parameter efficient finetuning methods. Adding reinforcement learning based search methods seems helping marginally w.r.t performance, training time. Also, why searching for parameters helps in image quality is not clear to me.
3. Comparison of reinforcement learning based search methods w.r.t grid search/ combinatorial search method would be required.
4. The rationale of using Eq.3 is not clear, why the authors choose to use power law method instead of any other combination?
5. The writing need to be improved. E.g., “Dreambooth” task is very weird, it should be called “personalized few-shot finetuning”. Overall, the motivation, method, experiments are not easy to follow.

**Questions:**

see weakness

---

> ### Author Response · Authors · 2023-11-21
>
> Thank you very much for your constructive feedback! We appreciate your recognition of the novelty of our methods. Regarding your concern, we would like to respond as follows.
>
> + Motivation for using RL: We aim to search architectures that have both low tuning cost and good performance, which is a multi-objective search task. The mainstream approach for solving this problem (multi-objective search)  mostly adopts the sample-eval-loop process based on reinforcement learning[1,2], so we followed their design. The power law in Eq.3 follows Mnas[2], which aims to balance the tuning performance and tuning cost, requiring a factor  for joint optimization objectives. In our revision, we will update the references and writing to help readers better understand our motivation.
>
>
> + Regarding the comparison with other search methods: We have supplemented our paper with a comparison between our method and evolutionary algorithm-based[3] and random search-based approaches[4], as shown in the table below.
>
> | Method                                          | Flowers102 | CUB-200 Birds | Food101 |
> |-------------------------------------------------|------------|---------------|---------|
> | LoRa                                            | 29.69      | 8.21          | 12.01   |
> | Random search [4]       | 47.21      | 23.42         | 17.82   |
> | Evolutionary search[3]         | 33.96      | 11.12         | 11.70   |
> | Ours                                            | **25.34**  | **6.63**      | **9.51**|
>
> We searched 1000 samples for all. As shown in the table, random search and evolution-based search failed to optimize an effective structure and fell behind the naive LORA method. This is because, in such a large search space, methods based on random walks or tournament selection cannot achieve effective exploration. We will include this comparison in our revised version.
>
>
>
> + Regarding why the visual quality will be enhanced, better structure can lead to more effective transfer, ensuring good transfer performance in the target domain and closer distribution distance (e.g., FID for image tasks). This has been proven in various parameter-efficient fine-tuning applications in NLP.  Additionally, our method introduces human performance scores as part of the multi-object reward to better align the tuning results with human preferences. As a result, the tuned results have better visual quality.
>
>
> + Regarding the writing. We appreciate your suggestions and will work on improving our writing.
>
>
>
> Finally, we thank you for your constructive feedback and recognition of our work again.
>
> [1] Zhou Y, Ebrahimi S, Arık S Ö, et al. Resource-efficient neural architect[J]. arXiv preprint arXiv:1806.07912, 2018.
>
> [2] Tan M, Chen B, Pang R, et al. Mnasnet: Platform-aware neural architecture search for mobile[C]//Proceedings of the IEEE/CVF conference on computer vision and pattern recognition. 2019: 2820-2828.
>
> [3]Real E, Moore S, Selle A, et al. Large-scale evolution of image classifiers[C]//International conference on machine learning. PMLR, 2017: 2902-2911.
>
> [4] Bergstra J, Bengio Y. Random search for hyper-parameter optimization[J]. Journal of machine learning research, 2012, 13(2).

---

> ### Comment · Reviewer_vpDh · 2023-11-22
> **Thank you for the response!**
>
> The rebuttal partially addressed my concerns. Although the method looks somewhat interesting, but more rigor need to be shown both in writing and in response (e.g., the authors mention [5], which is not present in rebuttal). Also, the paper could be updated here, which the authors haven't done. Therefore, I am keeping my rating.